# A Cross-Sectional Study on Pharmacy Students’ Career Choices in the Light of Saudi Vision 2030: Will Community Pharmacy Continue to Be the Most Promising, but Least Preferred, Sector?

**DOI:** 10.3390/ijerph18094589

**Published:** 2021-04-26

**Authors:** Dalia Almaghaslah, Abdulrhman Alsayari, Mona Almanasef, Amjad Asiri

**Affiliations:** 1Department of Clinical Pharmacy, King Khalid University, Abha 61421, Saudi Arabia; malmanasaef@kku.edu.sa (M.A.); amhamlan@kku.edu.sa (A.A.); 2Department of Pharmacognosy, King Khalid University, Abha 61421, Saudi Arabia; alsayari@kku.edu.sa

**Keywords:** career choice, pharmacy, Saudi Arabia, final year pharmacy students

## Abstract

Introduction: The Saudi Arabian healthcare divisions that recruit and hire pharmacists include hospital pharmacy, community pharmacies, universities, and research centres. Local studies showed that hospital pharmacy is the most preferred sector, while community pharmacy is the least preferred. However, jobs in hospital pharmacy are limited compared to community pharmacy. Hence, to accommodate the increasing numbers of pharmacy graduates and to facilitate the implementation of Saudi Vision 2030, which promotes primary healthcare and the participation of both private and non-governmental organisations in healthcare delivery, community pharmacy ought to be Saudised. This study was conducted to assess the career choices made by Saudi pharmacy students and the enablers that influence their career choice, especially in community pharmacy. Methods: A prospective cross-sectional approach was used. A total of 437 final year pharmacy students were recruited from 15 pharmacy schools around the country. Results: Salary and advancement opportunities as well as geographical location, benefits, and work environment were found to be “very important” enablers when making career decisions. Hospital pharmacy was selected as the most preferred sector by 242 (55.4%) of the participants, while community pharmacy was the least favoured pharmacy sector (17% or 6.2%). The enablers that might influence the consideration of a job in community pharmacy included career aspiration and social accountability. On the other hand, the barriers were personal beliefs about the sector and the nature of the work. Conclusions: The community pharmacy sector was found to be the least preferred sector to work in. The study revealed a list of enablers that the participants found to be relevant or of high relevance when choosing community pharmacy as a career pathway. Some of the enablers contribute to the role of the pharmacist towards the local community, social accountability, and towards the country’s Vision, such as interaction with the general public and educating them. Other enablers are related to the pharmacists’ career aspirations, such as owning a business. Some of the barriers that were found relevant include high workload, inflexible working hours, and limited opportunities for professional development. Localisation of community pharmacies would help to create more jobs for national pharmacists, increase the participation of female pharmacists in the workforce and support the achievement of Vision 2030. The barriers should be tackled on several levels: undergraduate curriculum, regulatory, and actual practice. Undergraduate education needs to include primary pharmaceutical care services in its curriculum. Regulatory changes include enforcing the renationalisation of the community pharmacy sector and permitting females to work in community pharmacies without location restrictions.

## 1. Introduction

Saudi Arabia is a high income and rapidly-developing nation in the Gulf region. According to the General Authority for Statistics, the estimated population reached nearly 33.4 million in early 2019, of which 37.8% were non-Saudi and 42.2% were females [1]. The Kingdom has been growing quickly in all divisions, including health [2].

The healthcare system in the country has significantly improved over the past several decades [3]. The system is comprised of two main sectors: government and private. Health services are provided at three levels: primary care through primary healthcare centres and community pharmacies, secondary care through public hospitals, and tertiary care through general or specialised hospitals [3,4]. The healthcare divisions that recruit pharmacists include hospital pharmacy, community pharmacies, universities, and research centres, the pharmaceutical industry, and organisations such as regulatory authorities [3].

Community pharmacy is regulated by the Ministry of Health. Owning a community pharmacy is restricted to Saudi-registered pharmacists, while working in a community pharmacy is restricted to licensed pharmacists regardless of nationality. The maximum number of community pharmacies a pharmacist can own is 30 [5]. They are usually run by one to two pharmacists or by a pharmacist and an assistant. They work between 8 and 12 h a day, 6 days a week [6]. The Saudi Commission for Health Specialties (SCFHS) is responsible for licensing and registering pharmacy graduates [3]. The Saudi Food and Drug Administration is responsible for regulating all pharmaceutical products, including over-the-counter products, prescribed medications, herbal products as well as food supplements [6].

The community pharmacy setting was the largest employment sector for pharmacists, employing 57% of the total workforce of 25,119. The second largest sector was scientific offices, factories, and drug stores, with 22% of employees in 2016. Hospital pharmacy was third, with 19% pharmacists, while primary healthcare centres employed only 476 (2%) pharmacists in 2016 [3,7].

In 2016, the Saudi government outlined a vision to transform the country, named Saudi Vision 2030. The Vision’s strategic goals involve creating more job opportunities by increasing the participation of private and non-governmental organisations, increasing the number of women in the job market, as well as helping young graduates make the best career choices [2]. In the pharmacy context, the Vision outlines is perfectly tailored to overcome the issues identified within the pharmacy workforce. The first issue is limited job opportunities in the government sector, which necessitates the participation of the private sector (community pharmacy) in providing jobs for national pharmacists. The second issue is that the number of female pharmacy students, which is slightly higher at 54.3% than their male counterparts, make up only 12.8% of the workforce [3]. The third issue is the gap between pharmacy students’ career preferences and the jobs available in the market. Saudi pharmacy students showed a strong preference for government hospital pharmacy jobs, but vacancies in this sector are scarce [3]. On the other hand, the largest employment sector, community pharmacy, was found to be the least favoured sector [8,9].

In 2017, a single institution study of Saudi female pharmacy students’ career choices showed that government hospitals were the preferred career pathway for (47, 50.9%) of respondents, followed by academia (21, 19.4%), while community pharmacy was the least favoured sector [8]. Another single institution study was conducted with final year Saudi pharmacy students. The study revealed that (41, 34%) of the participants chose hospital pharmacy for their future career, while (30, 25%) chose academia, (30, 25%) were not sure, (11, 9%) opted for owning a pharmacy and the same number (11, 9%) preferred working in the pharmaceutical industry [9]. Both of the studies were conducted in a single college of pharmacy, located in large, cosmopolitan, and multicultural cities, which may not provide generalisable findings. Moreover, the studies were conducted before the implementation of Saudi Vision 2030. Hence, a large-scale, multi-centre study was necessary to evaluate future pharmacy workforce career aspirations under the influence of Saudi Vision 2030. The Vision also encourages public–private partnerships in the delivery of preventative and primary care. Considering that community pharmacy is the first point of contact for patients and the role it played during the COVID-19 pandemic, especially with providing vaccinations, community pharmacy is an important sector in pharmacy practice that requires further attention.

We previously reported that the community pharmacy sector, the major employer of pharmacists in the country, is underutilised and employs a limited number of Saudi pharmacists; few females, in particular, are employed in this sector [3]. This study aimed to assess the career choices of Saudi pharmacy students and the enablers that influence their career choices, focusing on identifying the enablers, as well as the barriers, that influence Saudi pharmacists’ choices related to the community setting.

## 2. Material and Methods

### 2.1. Study Design

This study used a prospective, cross-sectional self-administered questionnaire, which was distributed among final year pharmacy students enrolled in 15 out of the 27 pharmacy schools in Saudi Arabia. The study was carried out from November 2019 to February 2020.

### 2.2. Sample Size and Sample Procedure

A convenience sample method was used. Saudi Arabia was divided into five regions, with several universities selected from each region. The universities are as follows: Southern region—King Khalid University, Jazan University, Najran University, and Al-Baha University; northern region—University of Tabuk; central region—King Saud University, Princess Noura University, and Al-Qassim University; eastern region—Imam Abdulrahaman bin Faisal University and King Faisal University; western region—Umm Al-Qura University, King Abdulaziz University, and Taibah University. For regions which have multiple universities—i.e., the central region, western region, and northern region—the selection of certain universities was based on the number of students. Universities with the largest number of students were recruited. On the other hand, for the other regions, which have fewer universities—i.e., the Southern and Eastern regions—all the universities were recruited. The sample size was based on the total number of final year pharmacy students in Saudi Arabia (approximately 2800) [3] then determined by using a Raosoft sample size calculator (Raosoft, Inc. Seattle, WA, USA; http://www.raosoft.com/samplesize.html, accessed on 15 November 2019) with a predetermined margin of error of 5% and a confidence level of 95%. The minimum sample size was determined to be 338 students. As indicated by Qualtrics (Qualtrics ^XM^, Seattle, WA, USA), the total case load was 501. A total of 437 students have provided a valid questionnaire response, and a total of 64 cases provided no response at all (just opened the questionnaire). Therefore, the response rate was (437/501) × 100 = 87.23%.

### 2.3. Participants

Final-year Saudi pharmacy students (fifth year of either the Pharm D or bachelor’s degree in Pharmacy) were targeted because they were close to graduation and were expected to join the job market within a few months. Exclusion criteria were currently employed, pharmacy student not ready for graduation and non-Saudi pharmacy student. Pharmacy colleges in Saudi Arabia offer either of the two entry-level degrees in pharmacy; Doctor of Pharmacy, Bachelor’s Degree in Pharmacy, or both. The Pharm D programme is more clinically focused and requires a full year of training compared to only 4 months for the Bachelor’s degree in Pharmacy [10]. However, both programmes are equivalent (in terms of registration) to a pharmacist at the Saudi Commission for Health Specialties.

### 2.4. Data Collection Tool

The structured questionnaire was developed from a variety of sources, including focus group discussions with final year pharmacy students at King Khalid University (two focus groups with 10 students each); the literature on factors affecting pharmacy career choices; previous studies on career perceptions [8,11]; discussions with four faculty members in the Clinical Pharmacy Department at King Khalid University, informal discussions with ten recent pharmacy graduates and five community pharmacists (who were invited to the campus for a two-hour workshop to discuss career choices, and the advantages and disadvantages of working in the community pharmacy sector) and the researchers’ personal experiences working as faculty members in the School of Pharmacy and being involved in community pharmacy services in the region. The questionnaire consisted of four domains: demographics/background information, general job considerations, career choices and factors supporting/discouraging work in the community pharmacy sector. The first section of the questionnaire gathered demographic information on the participants, such as age, gender, and relationship status. It also asked for the pharmacy school they attended. The second section assessed general job criteria, as developed by Ubaka and colleagues, as well as [12] using a five-point Likert scale: (1) very low importance; (2) low importance; (3) neutral; (4) high importance; and (5) very high importance. The third section of the questionnaire was adapted from a study done by Al Ghazzawi and colleagues [8]. In this section, students were asked to choose the pharmacy sector in which they would prefer to work after graduation: hospital, community, industry, academia/research, and other pharmacy sectors (e.g., regulatory affairs and medical supply). The last section of the questionnaire focused on the community pharmacy sector since it is the major employer in the country. The enablers related to community pharmacy were identified through the focus groups with students and discussions with community pharmacists. In this section, students were asked to choose a response (not relevant, low relevance, neutral, relevant, or high relevance) regarding the enablers and barriers related to working in the community pharmacy sector. The questionnaire was piloted with five final year pharmacy students at KKU who were representative of the study population, to determine the clarity of the language and the questionnaire structure. The results of the pilot study were not included in the results. The questionnaire was reviewed and modified based on the feedback received in the pilot. The data collection tool was finalised in the English language in the form of a self-administered web-based questionnaire. It was advertised via these social media channels (i.e., Twitter and WhatsApp) where that target sample gathered as a cohort, posts included a link to the questionnaire hosted on Qualtrics. The questionnaire link was accompanied by an introductory statement listing all of the inclusion criteria, i.e., being a final year pharmacy student in Saudi Arabia. A participant information sheet that involves detailed information about the study was included in the cover page of the questionnaire.

### 2.5. Data Analysis

The collected data were cleared, entered, and analysed by using the Statistical Package for Social Sciences (SPSS) (IBM, Chicago, IL, USA), version 24.0 for the Mac. Results were described in terms of frequencies. A Chi Square test was used to analyse demographic data, with a *p* value < 0.05 considered significant. Influential factors for general career consideration (6 items) used a Likert scale ranging from 1 (very unimportant) to 5 (very important); enablers (11 items) and barriers (15 items) for choosing the community pharmacy sector used a Likert scale ranging from 1 (not relevant) to 5 (high relevance). Scales items for enablers and barriers were categorised into themes. The distribution of the scale was presented in percentages, and using mean and SD. The internal consistency and reliability of the scales was assessed using Cronbach’s alpha coefficient, with the minimum recommended level being 0.70.

## 3. Results

### 3.1. Demographic Characteristics

A total of 437 final year pharmacy students participated in the study. Almost one-third (150, 34.3%) of them were males, while 287 (65.7%) were females (*p* = 0.0001).

Almost half of the responses were reported from the southern region (209, 47.8%) representing four pharmacy schools: King Khalid University, Jazan University, Al-Bahah University, and Njran University. The central region provided 64 (14.6 %) responses, representing King Saud University, Princess Noura University, and Al-Qassim University. Equal numbers of participants were from the eastern region and the western Region, with 55 (12.6%) responses each. These participants represented Immam Abdulrahman bin Faisal University, King Faisal University, King Abdulaziz University, Taibah University, and Umm al-Qura University. The northern region, represented by the University of Tabuk and Northern Borders University, had 54 (12.4 %) participants (*p* = 0.0001). Table 1 illustrates the background characteristics.

### 3.2. Influential Factors for General Career Consideration

The distribution of scale scores of the full set of six influential factors for general career consideration are presented in Table 2. Responses ranged from 1 (very unimportant) to 5 (very important). All items were skewed towards 5 (very important).

### 3.3. Career Choices

Hospital pharmacy was found to be the most preferred pharmacy sector by 242 (55.4%) of the participants, while academia/research and the pharmaceutical industry were the next favourite sectors, with 68 (15.6%) responses each. The pharmacy sector of regulatory affairs and medical supply was the second least favourite sector, with only 27 (3.9%) responses, followed by community pharmacy as the least favourite sector at 17 (6.2%) responses. A few participants opted to leave the profession (15, 3.4%) (*p* = 0.0001). These results are illustrated in Figure 1.

### 3.4. Community Pharmacy: Enablers/Barriers

The distribution of scale scores of the full set of 11 enablers for choosing the community pharmacy sector is presented in Table 3A. Responses ranged from 1 (not relevant) to 5 (high relevance). All items were skewed towards 5 (high relevance).

The distribution of scale scores of the full set of 15 barriers that were likely to prevent pharmacy students from choosing community pharmacy as a career is presented in Table 3B. Responses ranged from 1 (not relevant) to 5 (high relevance). All items were skewed towards “high relevance”.

Table 4 presents the distribution of barriers and enablers for choosing community pharmacy sector. These scales are treated continues variables ranging from 1 (not relevant) to 5 (high relevance). The mean value of the overall scale for enablers was 3.69 and for barriers was 3.53. All scales had Cronbach alpha coefficient greater than 0.7 indicating inter-item reliability.

## 4. Discussion

The current study assessed the career choices of Saudi pharmacy students, evaluating the factors that influence their career choices. Based on previous study findings, one focus of this research was the community pharmacy sector, as this healthcare provider is likely to play a key role in achieving Vision 2030 [2,8,9,13,14,15]. It focused on identifying the factors, both positive and negative, that inform Saudi pharmacists’ decisions about working in the community setting.

The pre-graduates’ job considerations showed that salary and advancement opportunities were very important factors in making a job choice, the same factors that were highlighted in previous studies [11,16]. Less important factors included benefits, geographical location, work environment, and flexible work schedule. This survey found that hospital pharmacy was the most preferred career pathway, while community pharmacy was the least preferred sector. The same findings were reported in two single-centre studies in Saudi Arabia [8,9]. By contrast, international studies listed community pharmacy as one of the most preferred career choices in Australia, the United Kingdom, the United States, Nigeria, Syria, Japan, and Malaysia [11,12,16,17,18,19,20].

The study revealed a list of enablers that the participants found to be relevant when choosing community pharmacy as a career pathway. Some of the enablers contribute to the role of the pharmacist towards the local community, social accountability, and towards the country’s Vision, such as interaction with the general public, educating them and changing their negative perceptions of the sector. Other enablers were related to the pharmacist career aspiration, such as owning a business, developing new skills such as soft skills, diagnostic skills of minor ailments, counselling skills and widening their knowledge on certain pharmaceutical and non-pharmaceutical products.

One of the enablers is the fact that job opportunities in the most preferred sector, hospital pharmacy, are very limited.

Some of the barriers that future pharmacists found to be relevant were related to their readiness to work in community pharmacy, i.e., previous training as undergraduate students. Other barriers found to be relevant were related to the nature of the work in this career pathway, i.e., high workload, inflexible working hours, limited professional development opportunities, low job security, concerns about personal safety, low salaries, and being profit-oriented. Other barriers, such as the country’s regulations and employers’ preference for hiring non-Saudi pharmacists, were also found to be relevant, although they are not directly related to the sector itself, but rather are personal perceptions of the sector. Cultural constraints such as working with the opposite sex, however, were found to be irrelevant.

Some of the barriers which were found relevant—including high workload, inflexible working hours, and limited professional development opportunities—concur with the negative factors previously reported in other studies [11,12].

The study participants listed a lack of previous training as one of the barriers that prevented them from choosing community pharmacy. Pharmacy schools should assume responsibility for preparing their graduates to meet the needs of the market. The curriculum in pharmacy schools should, therefore, be designed to focus on primary healthcare and preventative medicine, along with therapeutic care. The curriculum should also equip students with the skills required for working in community settings, such as communication skills, patient counselling skills, information technology and marketing skills [21,22,23].

Furthermore, pharmacy schools should motivate young pharmacists to be socially accountable for educating society, especially through working in ‘front line’ healthcare facilities. By gaining these values and skills, future pharmacists would be encouraged to join a community setting. Additionally, career advisory services in Saudi universities should play a role in increasing pharmacy students’ awareness of possible employment in this sector.

The limited pharmaceutical care services—i.e., clinical roles of the community pharmacists other than dispensing—was one of the barriers related to work in community pharmacies. The practice of pharmaceutical care in a community pharmacy is limited, with medication dispensing being the dominant practice [5]. One study found that community pharmacies are commercial and profit-centred, with limited patient-focused pharmaceutical care [24].

Another study highlighted several deficiencies in the practices of community pharmacists, including poor knowledge of drug-drug interactions, poor adherence to the profession’s legislative acts regarding the dispensing of certain medications, lack of continuous professional development and little-to-no patient counselling [25]. Widening the scope of pharmaceutical services offered by community pharmacists would expand their role in this sector, hence creating more jobs. However, to achieve this, a needs-based education approach that focuses on primary/preventative pharmaceutical care services should be adopted by undergraduate pharmacy programmes around the country.

Interestingly, “perceptions of an employer’s preference for hiring non-Saudi pharmacists” was among the barriers. The International Labour Organization reported that private organisations in the Gulf region tend to hire non-indigenous employees, as they are more flexible regarding workload and working hours, as well as being a lower labour cost [26]. In this scenario, creating more jobs for national pharmacists in the private sector would require the government to enforce its law regarding the re-nationalisation of the profession, particularly increasing the number of local pharmacy schools and pharmacy graduates.

Surprisingly, “cultural constraints” and “working with the opposite sex” were not found to be barriers that prevent future pharmacists from working in community environments. In contrast, the study by Al Ghazzawi and colleagues reported that the conservative Saudi culture prevented female pharmacists from working in certain jobs [8]. However, the recent changes in the country, i.e., the shift to moderate Islam as well as women’s empowerment, have probably changed participants’ social perceptions of career choices [2].

Another interesting finding is that participants found society’s negative perception of community pharmacy among the barriers, as previously reported in the study by Al-Asfour [13].

Participants found that the country’s regulation that women are not allowed to work in community pharmacies unless they are linked to private hospitals or located in shopping malls to be a very important barrier in choosing the community sector. This is probably because 65.7% of the sample was female.

The findings of this study suggested overcoming the identified barriers should be done at several levels: education by incorporation of community pharmacy focused education and training into undergraduate curriculum. Regulatory by imposing the Saudisation of the community pharmacy sector and allowing females to work in community pharmacies without location restrictions. Actual practice by expanding the role community pharmacists provide as the first point of contact for patients as well as fulfilling the role of community pharmacy as a sector in the public–private partnership that the government is seeking to cover primary and preventative care.

The study had some limitations. The sample included limited variability, since almost half of the participants were from the southern region and only one-third were males. Another limitation was that providing a comparison between this study’s findings and those of international studies was not possible, due to country-specific factors that were not applicable to other cultures. The study design, quantitative in nature, did not allow in-depth understanding of students’ perceptions of the enablers and barriers. Hence, future work using a qualitative study design should be considered, to elaborate on the reasons behind their choices and attitudes, and to further explain some of the identified phenomena, e.g., gender disparities.

## 5. Conclusions

A major focus of this study was identifying the enablers and barriers that influence Saudi pharmacists’ decisions regarding work in the community setting. Community pharmacy, the major employer, was the least preferred sector of respondents, and several barriers were identified. These barriers should be tackled on several levels: undergraduate curriculum, regulatory, and actual practice. Undergraduate education needs to include primary pharmaceutical care services in its curriculum. Regulatory changes include enforcing the renationalisation of the community pharmacy sector and permitting females to work in community pharmacies without location restrictions. At the practical level, the community pharmacy sector should be encouraged to expand the pharmaceutical services it provides. Overcoming these barriers would create more jobs for the increasing numbers of pharmacy graduates and help in achieving Saudi Vision 2030. Further research focusing on the role of community pharmacy in fulfilling public–private partnerships for effective delivery of preventative primary healthcare in Vision 2030 is essential.

## Figures and Tables

**Figure 1 ijerph-18-04589-f001:**
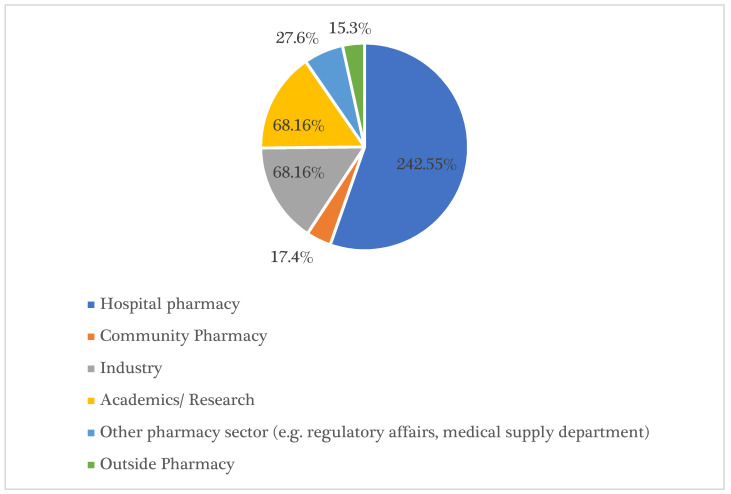
Pharmacy Students’ Career Goals after Graduation.

**Table 1 ijerph-18-04589-t001:** Demographic characteristics.

		***n* (%)**		***p* Value**
Gender				
Male		150 (34.3)		0.0001 *
Female		287(65.7)		
Age				
<21		54 (12.4)		0.001 *
21–23		328 (75)		
<23		55 (12.6)		
Geographical Location	University	*N* (%)	Total (%)	
Southern region	King Khalid University	131(22.4)	209 (47.8)	0.0001 *
	Jazan University	59 (10.1)		
	Al-Baha University	17 (2.9)		
	Najran University	6 (1)		
Northern region	University of Tabuk	26 (4.5)	54 (12.4)	
	Norther Borders University	26 (4.5)		
Central region	King Saud University	26 (4.5)	64 (14.6)	
	Princess Noura University	24 (4.1)		
	Al-Qassim University	15 (2.6)		
Western region	King Abdulaziz University	14 (2.4)	55 (12.6)	
	Umm Al-Qura University	25 (4.3)		
	Taibah University	16 (2.7)		
Eastern region	King Faisal University	25 (4.3)	55 (12.6)	
	Imam Abdulrahman bin Faisal	27 (4.6)		

Chi square test, * *p* < 0.05.

**Table 2 ijerph-18-04589-t002:** Distribution of influential factors for general career consideration ranging from 1 (very unimportant) to 5 (very important).

	*n*	Distribution of Responses (%)					Skew	Mean	SD
Factor Description		1	2	3	4	5			
1. Benefits	424	1.2	2.4	7.1	41.3	48.1	−1.49	4.33	0.8
2. Salary	425	4	8.5	17.4	34.6	35.5	−0.88	3.89	1.1
3. Geographical location	425	2.8	4.2	12.7	38.8	41.4	−1.24	4.11	0.97
4. Work environment	423	1.4	2.8	10.2	36.4	49.2	−1.38	4.29	0.87
5. Advancement opportunities	425	3.5	8.7	21.9	32.5	33.4	−0.72	3.83	1.09
6. Flexible work schedule	416	6.5	16.3	16.3	4.1	56.7	−0.72	3.88	1.39

**Table 3 ijerph-18-04589-t003:** (**A**) Distribution of enablers for choosing the Community Pharmacy Sector ranging from 1 (not relevant) to 5 (high relevance). (**B**) Distribution of barriers for choosing the Community Pharmacy Sector ranging from 1 (not relevant) to 5 (high relevance).

(**A**)									
**Enabler Description**		**Distribution of Responses (%)**					**Skew**	**Mean**	**SD**
	***N***	**1**	**2**	**3**	**4**	**5**			
1. Limited work opportunities in other sectors	421	10	16.2	22.8	35.4	15.7	−0.41	3.31	1.2
2. Daily interaction with general public	421	4.3	8.3	18.8	38.5	30.2	−0.84	3.82	1.08
3. Opportunity to educate society	422	5.5	8.5	16.4	37.4	32.2	−0.803	3.82	1.14
4. Opportunity to change society negative perception of this sector	422	5.5	8.5	16.4	37.4	32.2	−0.803	3.82	1.14
5. Opportunity to improve soft skills, such as communication skills, marketing skills and IT skills	422	11.1	13	22.3	25.8	27.7	−0.45	3.46	1.32
6. Desire to own a business	421	7.4	5.7	18.3	40.4	28.3	−0.96	3.76	1.14
7. Recognising and treating minor aliments by recommending over-the-counter medicines	421	4.5	5.7	13.1	40.1	36.6	−1.16	3.99	1.06
8. Counselling opportunities for prescribed medications	419	8.4	10	22.4	30.5	28.6	−0.63	3.61	1.23
9. Opportunity to learn more about non-pharmaceutical products, such as cosmetics and dietary supplements 420	420	6.4	8.8	18.1	31.2	35.5	−0.836	3.8	1.19
10. Saudising this sector—achieving Saudi Vision 2030	419	7.6	9.3	20	34.1	28.9	−0.74	3.67	1.2
11. Benefits, including health insurance	416	8.2	10.3	21.9	31.3	28.4	−0.637	3.61	1.23
(**B**)									
**Barrier Description**		**Distribution of Responses (%)**					**Skew**	**Mean**	**SD**
	***N***	**1**	**2**	**3**	**4**	**5**			
1. Lack of pervious training during pharmacy programmes, which makes working independently challenging	415	5.5	8.2	25.5	31.1	29.6	−0.66	3.71	1.14
2. Weak pharmaceutical care services in this sector	415	14.7	8	30.1	19.8	27.5	−0.39	3.37	1.35
3. Country’s regulations	415	7.2	5.1	19.3	26.3	42.2	−0.99	3.91	1.21
4. Perceptions of an employer preference (majority prefer non-Saudi pharmacists)	416	5.8	8.2	30.3	26.4	29.3	−0.53	3.65	1.15
5. It is a profit-oriented sector	415	29.2	12.3	28.2	14.7	15.7	0.15	2.75	1.42
6. Cultural constraints	415	34.5	16.9	27	11	10	0.45	2.47	1.34
7. Working with the opposite sex	416	17.5	13.5	16.3	21.9	30.8	−0.37	3.35	1.47
8. Society’s negative perceptions (perceive community pharmacy as supermarket)	416	14.9	10.6	22.1	22.6	29.8	−0.44	3.42	1.4
9. Feeling over-qualified for the job	415	8.7	9.4	22.7	29.2	30.1	−0.65	3.63	1.24
10. Low job security	415	11.8	8	29.9	25.5	24.8	−0.48	3.44	1.27
11. Concern about personal safety	412	7.5	7	19.9	31.3	34.2	−0.84	3.78	1.21
12. Limited professional development opportunities	417	5.3	6	19.9	30.9	37.9	−0.93	3.9	1.13
13. Low salaries compared to other sectors	415	8.2	7.2	28.7	28.2	27.7	−0.59	3.6	1.2
14. High workload	418	6.5	7.9	17	27.5	41.1	−0.93	3.89	1.21
15. Irregular working hours	418	6.5	7.9	17	27.5	41.1	−0.93	3.89	1.21

**Table 4 ijerph-18-04589-t004:** Distribution and internal consistency of barriers and enablers for choosing community pharmacy sector scale.

Description of Scale	*n*	Distribution of Responses (%)					Skew	Mean	SD	Cronbach α
		≤1	≤2	≤3	≤4	≤5				
Enablers										
Career aspiration (Items 2, 5, 6, 9, 11)	413	1.5	5.3	21.1	64.9	100	−0.9	3.69	0.83	0.74
Social accountability (Items 1, 3, 4, 7, 8, 10)	417	1.2	7	20.6	63.3	100	−0.94	3.7	0.88	0.85
Overall scale (All items)	411	0.7	5.6	18.7	62.8	100	−1.02	3.69	0.81	0.89
Barriers										
Cultural factors (Items 6, 7)	414	13.5	28.5	64	87.9	100	−0.03	2.09	1.15	0.5
Personal beliefs (1, 4, 8, 9)	412	1.5	4.4	27.9	75.5	100	−0.49	3.61	0.84	0.6
Nature of work (Items 2, 3, 5, 10, 11, 12, 13, 14, 15)	408	0.7	4.4	21.8	67.2	100	−0.71	3.63	0.79	0.81
Overall scale (All items)	404	0.5	4	23	76	100	−0.61	3.53	0.74	0.86

## Data Availability

The data presented in this study are available on request from the corresponding author.

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
