# Peer review of "A Cross-Sectional Study on Pharmacy Students’ Career Choices in the Light of Saudi Vision 2030: Will Community Pharmacy Continue to Be the Most Promising, but Least Preferred, Sector?"

_ijerph, 2021, doi:10.3390/ijerph18094589_

Round 1

Reviewer 1 Report

In the new revised version, the article has more clearly outlined goals, research, analytical methods and conclusions.

Author Response

Thank you for your positive response 

Reviewer 2 Report

Thank you to the authors for taking previous feedback on board to substantially revise the work, which is now much stronger and better suited to international publication. The methods are better described, the findings clearer and there is a clearer discussion of what they mean and why it matters. There are a few more areas where the authors might consider further strengthening the paper to really reinforce their work: 

  • End of introduction - there is a clear link between the previous study and this one but the end of the introduction should spotlight why THIS paper is different and building better on the previous, and why it's needed and now
  • Section 2.2 - something is missing here from the sampling description; it is clear that there is a convenience sample drawn from students from the selected universities, but how were participants approached/informed about the study? Were the cohorts contacted as a whole via the university (e.g. emailed or informed on learning platforms by their program administrators)? Did the authors contact these cohorts through other means (e.g. social media groups in which they are gathered as a cohort)? 2.4 would seem to suggest the authors recruited their participants via social media (Twitter and WhatsApp) but in that case, how was the recruitment targeted (and confined) to the specified cohorts? It is good that the minimum threshold of the sample is detailed here, but the reader should find out what the actual number of participants was here also.
  • 2.4 - can the authors clarify what they mean by the data collection tool being "a self-administered web-based questionnaire that was conducted via social media (i.e. Twitter and WhatsApp)"? Is it the case the survey was advertised/announced via these social media channels and posts included a link to the questionnaire hosted on another platform/website? Surely the questionnaire was not completed within the Twitter/WhatsApp apps themselves?
  • Section 3.4, Figure 1: I am unsure about the pie chart representation. Was the questionnaire instrument designed such that the respondents could only choose one option for the question "career goals after graduation"? If not then a pie chart is not an accurate representation as you cannot really be showing 100% effectively when participants are choosing multiple answers (Some might pick 1, others 3, so then the numbers aren't equivalent either to the total participant numbers or the strength of preferences.). It would suit if only one option was allowed to be chosen (if so, specify) but then present % only, detail full numbers in the text. 
  • 4 - Discussion: this really should lead with a quick strong summary of the main findings/contributions of the study, rather than more background. The existing first and second paragraphs just repeat material from the introduction - it takes until Paragraph 6 to see "this study revealed...": this should be the 'launching pad' of the discussion really highlighting that the study and paper contribute new important data. Several other paragraphs lead with findings from other studies or descriptive information about the country and sector; this material is good but should be used to support statements of findings from THIS study, rather than leading the argument.
  • Line 388-390 - re participants feeling neutral about regulations over women's community pharmacy work; the authors suggest here a link with this finding and the 1/3 of male participants, but in what way? Is the suggestion that women (the 2/3 of participants) are the ones who feel neutral about it and if there were more men it would be different? Or that the minority (1/3 male) is more influential than the majority (2/3 female) in shaping this finding?

Author Response

Comment

Reply

Thank you to the authors for taking previous feedback on board to substantially revise the work, which is now much stronger and better suited to international publication. The methods are better described, the findings clearer and there is a clearer discussion of what they mean and why it matters. There are a few more areas where the authors might consider further strengthening the paper to really reinforce their work

Thank you for your time and effort in reviewing our work.

End of introduction - there is a clear link between the previous study and this one but the end of the introduction should spotlight why THIS paper is different and building better on the previous, and why it's needed and now

We have added this section based on your suggestion in first round of review. We are not sure what else we can add.

 Both of the studies were conducted in a single college of pharmacy, located in large, cosmopolitan and multicultural cities, which may not provide generalisable findings. Also, the studies were conducted before the implementation of Saudi Vision 2030. Hence, a large-scale, multi-centre study was necessary to evaluate future pharmacy workforce career aspirations under the influence of Saudi Vision 2030. The Vision also encourages public-private partnerships in the delivery of preventative and primary care. Considering that community pharmacy is the first point of contact for patients and the role it played during the COVID-19 pandemic, especially with providing vaccinations, community pharmacy is an important sector in pharmacy practice that requires further attention.   

Section 2.2 - something is missing here from the sampling description; it is clear that there is a convenience sample drawn from students from the selected universities, but how were participants approached/informed about the study? Were the cohorts contacted as a whole via the university (e.g. emailed or informed on learning platforms by their program administrators)? Did the authors contact these cohorts through other means (e.g. social media groups in which they are gathered as a cohort)?

We contacted these cohorts through social media groups in which they are gathered as a cohort

2.4 would seem to suggest the authors recruited their participants via social media (Twitter and WhatsApp) but in that case, how was the recruitment targeted (and confined) to the specified cohorts? It is good that the minimum threshold of the sample is detailed here, but the reader should find out what the actual number of participants was here also.

The questionnaire link was accompanied by an introductory statement listing all of the inclusion criteria, i.e., being a final year pharmacy student in Saudi Arabia. A participant information sheet that involves detailed information about the study was included in the cover page of the questionnaire.

2.4 - can the authors clarify what they mean by the data collection tool being "a self-administered web-based questionnaire that was conducted via social media (i.e. Twitter and WhatsApp)"? Is it the case the survey was advertised/announced via these social media channels and posts included a link to the questionnaire hosted on another platform/website? Surely the questionnaire was not completed within the Twitter/WhatsApp apps themselves?

Thank you for your valid comment.

What we mean by (self-administered) that participants filled the questionnaire by themselves without an interviewer, and (web-based) through Qualtrics.

That is correct the survey was advertised/announced via these social media channels and posts included a link to the questionnaire hosted on another platform/website (Qualtrics).

Section 3.4, Figure 1: I am unsure about the pie chart representation. Was the questionnaire instrument designed such that the respondents could only choose one option for the question "career goals after graduation"? If not then a pie chart is not an accurate representation as you cannot really be showing 100% effectively when participants are choosing multiple answers (Some might pick 1, others 3, so then the numbers aren't equivalent either to the total participant numbers or the strength of preferences.). It would suit if only one option was allowed to be chosen (if so, specify) but then present % only, detail full numbers in the text

The question was a multiple-choice question where participants can choose only one answer. The data was presented in a table, but the academic editor requested chaining it to a pie chart.

4 - Discussion: this really should lead with a quick strong summary of the main findings/contributions of the study, rather than more background. The existing first and second paragraphs just repeat material from the introduction - it takes until Paragraph 6 to see "this study revealed...": this should be the 'launching pad' of the discussion really highlighting that the study and paper contribute new important data. Several other paragraphs lead with findings from other studies or descriptive information about the country and sector; this material is good but should be used to support statements of findings from THIS study, rather than leading the argument

This point was also raised by the other reviewer who suggested deleting the first few paragraphs of the discussion, so we deleted them. 

Line 388-390 - re participants feeling neutral about regulations over women's community pharmacy work; the authors suggest here a link with this finding and the 1/3 of male participants, but in what way? Is the suggestion that women (the 2/3 of participants) are the ones who feel neutral about it and if there were more men it would be different? Or that the minority (1/3 male) is more influential than the majority (2/3 female) in shaping this finding?

This is not correct statement. After conducting scale analysis as the academic editor suggested the mean of this barrier was 3.91 indicating this barrier is very important. So it does make sense as 2/3 of participants are females.

The statement was corrected as follow:

Participants found that the country's regulation that women are not allowed to work in community pharmacies unless they are linked to private hospitals or located in shopping malls to be very important barrier for choosing community sector. This is probably because 65.7% of the sample was female.

Reviewer 3 Report

Overall I thought this was a good effort.  Your background and introduction were well done.   The methodology was mostly good.  You did not say how you were able to contact your study participants.  Did you have their email addresses?  If so, how were you able to get them from so many different schools.  You mentioned the use of social media.  Does that mean you did not have a list of names and emails to send your survey to?  This should be clarified.  

In the results you indicate 437 study participants.  What was your response rate?  

Your discussion should focus on your results and interpretation of those results.  I found much of your discussion in the first 6 paragraphs to be about previous studies.  You can either eliminate these paragraphs or re-write them so the focus is on your results.  

Author Response

Comment

Reply

Overall I thought this was a good effort.  Your background and introduction were well done.   The methodology was mostly good. 

Thank you for your time and effort in reviewing our work.

You did not say how you were able to contact your study participants.  Did you have their email addresses?  If so, how were you able to get them from so many different schools.  You mentioned the use of social media.  Does that mean you did not have a list of names and emails to send your survey to?  This should be clarified. 

This point was clarified as follow

In the results you indicate 437 study participants.  What was your response rate?

Your discussion should focus on your results and interpretation of those results.  I found much of your discussion in the first 6 paragraphs to be about previous studies.  You can either eliminate these paragraphs or re-write them so the focus is on your results. 

The paragraphs were deleted

Round 2

Reviewer 2 Report

The authors should be commended for their rigorous efforts in pursuing this paper through multiple rounds of peer review, and for engaging with every piece of feedback with openness and integrity. Every previous issue that was raised in my previous reviews has been amply addressed in this most recent revision, and this paper is now both scientifically robust and clear and engaging to read. I have no hesitation in recommending publication of this paper by the journal, and thank the authors for their commitment. 

This manuscript is a resubmission of an earlier submission. The following is a list of the peer review reports and author responses from that submission.

Round 1

Reviewer 1 Report

The article deals with the education of students of pharmacy and future job opportunities in the following sectors: hospital pharmacy, community pharmacies, universities and research centers. According to Saudi 2030 Vision community pharmacy ought to be Saudised.  This goal is hard to achieve without the right staff. This research was carried out on a group of last-year students of pharmacy, which shows that the largest employment sector, community pharmacy, was found to be the least favorite sector. The research has some limitations, mentioned by the authors themselves - a small number of men compared to women and overrepresentation of selected universities from the south of the country. Nevertheless, the advantage of these studies, in contrast to previous studies, is the large number of people surveyed and the fact that they are cross-sectional studies carried out in different regions of the country (in 15 out of the 27 pharmacy schools in Saudi Arabia).The article presents numerical data on pharmacies in individual sectors and the number of employees. Legal restrictions were presented. The 2030 Vision targets are presented in relation to the pharmacies sector.However, the research focused on preparing students for work in community pharmacy and promoting this form of work.

During the study, the students' career choices were determined.The authors mention the factors that influence students career choices: facilitating factors and the barriers that influence Saudi pharmacists' choices related to the community setting.The authors clearly list the basic barriers and factors attracting students to this form of work and determine what social factors influence the perception of work in community pharmacies.

The article is of a practical nature. Its main goal is to indicate guidelines for the implementation of government plans. The barriers indicated by the respondents reflect the specificity of Saudi reality, local law, social conditions and the labor market. Therefore, these results can only be used or compared in countries with similar specificities.The results of the study can be used as one of the elements of the description of the situation in the medical services sector.The reported conclusions are a task for the Saudi government and should be taken into account in legal, organizational and promotional changes among the community if 2030 Vision is to be implemented.

Finally, it should be noted that the preferences of students regarding the choice of the future workplace are often verified by market needs. The barriers that students write about are the result of their general beliefs and not of specific experience. An important supplement to the study would be to examine the current employees of community pharmacies in terms of real (not anticipated or imagined) barriers to the development of their activities.It should be emphasized that in terms of methodology, the study was well conducted. Its results are practically applicable, and can also be used to develop labor market incentives or a special educational path. The article may also be used as a basis for designing special educational programs for future and already working English-speaking pharmacists (e.g. Arabic language classes). The 2030 Vision goal should definitely be achieved by influencing an already functioning market through incentives rather than constraints. This means ready-made jobs for students to apply for. The conclusions could be extended with a proposal for changes or research in this area.  Any comparisons between countries must take into account macroeconomic and social constraints. The article may also be used in the future as a reference point for research on this topic.Minor editorial errors were noticed in the article, e.g. on page 1 a double space, on page two - no opening brackets. Therefore, the article requires re-reading and correction.

Reviewer 2 Report

This is a very well prepared submission. My only thought is that you could add some more detailed recommendations in the Abstract.

Reviewer 3 Report

Dear sirs,

I have read your manuscript with interest. Unfortunately, I cannot endorse its publication in a journal like IEJHP for various reasons:
1.- The paper has an exploratory nature and, therefore, descriptive. In a scientific journal, it is expected that empirical works try to have an explanatory character.
2.- Given that the objective is descriptive, the research problem's formulation would be of no interest. For this reason, the authors do not discuss in depth the literature on Human resources that might be relevant.
3.- The statistical methods are bivariate, so the interrelation between the variables is not addressed. To conclude that the study has an explanatory character, a longitudinal study would have to have been carried out, and multivariate models had to be formulated.

For all the above, unfortunately, I am forced to reject the paper.

Reviewer 4 Report

Review: Community Pharmacy—The Most Promising but Least Preferred Sector: Its Role in Fulfilling The Saudi Healthcare Plan in the 2030 Vision

Thank you for the opportunity to review your paper. The topic is interesting and there is a contribution it can make towards literature in support of health system research in primary care and pharmacy care, but the paper needs significant revision in order to clarify and sharpen the argument it is making. Some specific feedback below:

1:

  • Page 2 Paragraph 3:Three issues are listed which are identified as relevant to the pharmacy workforce. The second and third issues are clearly articulated, but the first gets lost in the third sentence of the parargraph. Instead of the colon after “identified within the pharmacy workforce”, full stop, and begin a new sentence “The first issue is that there are limited job opportunities…”
  • Page 2 Paragraph 3: “the number of female pharmacy students, which is slightly higher (54.3)% than their male counterparts, make up only 12.8% of the workforce “ – It would be worth addressing briefly why – do they take roles in sectors outside pharmacy? Do they turn to home duties? It might be reasons unknown, but perhaps this should be stated then.
  • Page 2 Paragraph 4: the presentation of numbers here reads awkwardly. Suggest just using the percentages with no parentheses () rather than the both the percentage and raw numbers, as this section seems to be identifying trends.
  • In the final two paragraphs, previous studies are listed. Be sure to state very clearly how this current study is different, and why it is important to study this again now and in this way.
  • One thing missing from this introduction is a justification WHY community pharmacy matters. Why does attention need to be paid to moving students towards the community sector rather than the private? Is it merely a case of employability? Or has it been evident (perhaps locally or internationally; was it the case in the recent COVID-19 pandemic particularly, that might be the perfect reason?) that shifting the burden of pharmaceutical care (and the bulk of staff) to the community sector is the best move for the society?

2:

  • 3 the final two sentences should be combined into one.
  • 4 ref 10 is out of place
  • 4 there seems to be a discrepancy between the purposive sample and the recruitment method; if the sample is restricted to set cohorts in set institutions, can you explain the decision to recruit via open social media rather than a targeted recruitment distributed direct to these cohorts with the assistance of the institutions? Need to justify the appropriateness of this approach.
  • 5 more clearly explain the purposes of your tests. What type of Chi square are you conducting and what are they determining? What is your Kruskal Wallis H test is looking for; more than just the “pharmacists’ attitudes and practices”, what are you looking for these to differ based on? NB the statistical methods used should be mentioned in the abstract so they are immediately apparent.

3:

  • I cannot comment much on the findings of section 3 as I am not a quantitative/statistical analysis specialist. However, in Table 4 the item “Daily interaction with the general public” has a P value of 0.03; based on the decision *p<0.05 for your Kruskal Wallis shouldn’t this be significant?
  • Consider splitting table 4 into two defined and distinct tables (4a and 4b) rather than one long list.

4:

  • This section should really lead with a short summary (1-2 sentences) of the key findings of 4 that are at the heart of the argument in the paper. The study of this paper was not mentioned until paragraph 6, so these original findings were lost in this discussion.
  • Page 7 paragraph 5: “At the moment, few of the international staff speak Arabic, while the most of them speak English as a second language”. Is there a reference for this? Additionally, this is linked to the paragraph before and the idea that pharmacists who speak the national language may be better for cultural sensitivity; perhaps this sentence should move up to the previous paragraph?
  • Page 7 paragraph 7: can you explain this difference between your findings and the international findings? What factors specific to SA might affect the career choice so strongly? Alternatively, if it is not possible to tell given the scope of this study, mention this as a limitation or avenue for further research.
  • Page 8 paragraph 3 –“ Countries regulation and employer preference to hire non-Saudi pharmacists” – this is a little unclear; does it mean that there is no regulations requiring companies to hire Saudi, or regulations allowing them to hire preferentially? Additionally, can you provide data that there is a real trend of preference to hire non-Saudi, or should this read “perception of an employer preference…”?
  • Page 8 paragraphs 5 and 6: here there is the beginning of some ideas for what pharmacy schools should do to begin remedying some of these factors/barriers and improve the matriculation of students to the community sector. A good, strong way to end this discussion section would be to tease out more broadly what are some of the other things that different stakeholders should do to aid this matriculation; some of the other stakeholders mentioned are the government, employers, etc. Some of these are touched on, but a clear and strong “next steps” paragraph would answer the question “So What?” about these findings, what can be done to solve the problems they highlight
  • Appreciate that Limitations are mentioned, but these are only brief. Consider what this study design, particularly its quantitative nature, could not tell you. What future/further research (particularly qualitative) could shed more light on what these barriers mean for students, where they get their attitudes from, and what lies behind some of the phenomena (e.g. gender disparities) identified.

Throughout:

  • Minor: formatting consistency, put endnote numbers after the punctuation mark of the sentence.
  • There are multiple expression errors, missing words, and errors with spacing and punctuation. Needs a careful read-through, maybe by a third party beyond the author team, using fresh eyes and an eye to the fine detail of the language expression
  • There are multiple errors in referencing where there is a blend of in-text and numbered styles; be sure to conform to the MDPI referencing style (numbers only).